# Association between Sleep Duration and Incident Diabetes Mellitus in Healthy Subjects: A 14-Year Longitudinal Cohort Study

**DOI:** 10.3390/jcm12082899

**Published:** 2023-04-16

**Authors:** Jin ha Jang, Wonjin Kim, Jin Sil Moon, Eun Roh, Jun Goo Kang, Seong Jin Lee, Sung-Hee Ihm, Ji Hye Huh

**Affiliations:** 1Department of Internal Medicine, Hallym University Sacred Heart Hospital, Anyang 14068, Republic of Korea; 2Division of Endocrinology and Metabolism, Department of Internal Medicine, Gangnam CHA Medical Center, CHA University School of Medicine, Seoul 06135, Republic of Korea; 3Department of Biostatistics, Yonsei University College of Medicine, Seoul 06273, Republic of Korea; 4Division of Endocrinology and Metabolism, Department of Internal Medicine, Hallym University Sacred Heart Hospital, Anyang 14068, Republic of Korea

**Keywords:** sleep duration, sleep quality, excessive daytime sleepiness, diabetes, insulin secretion

## Abstract

Background: This study aimed to investigate whether sleep duration and/or quality are associated with incident diabetes mellitus (DM). Methods: A total of 8816 of 10,030 healthy participants were enrolled in a prospective cohort study. Sleep duration and quality questionnaires were completed. Sleep quality was assessed using the Epworth Sleepiness Scale (ESS), which measures excessive daytime sleepiness in individuals. Results: During the 14-year follow-up period, 18% (1630/8816) were diagnosed with DM. A U-shaped relationship was observed between sleep duration and incident DM, with the highest risk observed when sleep duration was ≥10 h/day (hazard ratios (HR) 1.65 [1.25–2.17]). This group exhibited decreased insulin glycogenic index, a marker of insulin secretory function, during the study period. Among study participants who slept less than 10 h/day, the risk of incident DM increased when the ESS score was >10. Conclusions: We found that the association between sleep duration and incident DM was U-shaped; both short (≤5 h) and long (≥10 h) sleep durations were associated with an increased risk for the occurrence of incident DM. When sleep duration was 10 h or longer per day, there was a tendency to develop DM due to decreased insulin secretory function.

## 1. Introduction

In recent decades, the prevalence of diabetes has increased worldwide [1]. With an estimated 1.5 million deaths directly attributable to diabetes in 2019, diabetes was the ninth leading cause of death [2]. From the report of the International Diabetes Federation [3], approximately 537 million adults have been diagnosed with diabetes, and the health burden has increased by 316% in the last 15 years. The complications of diabetes and the disease itself have significant public health implications. Given this significant burden, it is critical to identify modifiable lifestyle factors associated with lower diabetes risk in clinical practice.

Sleep is a biobehavioral phenomenon that is regulated by circadian, homeostatic, and neurohormonal processes [4]. Most people spend approximately one-third of their lifetime sleeping, and it is an important lifestyle factor in individuals. Inadequate sleep and several key features of obstructive sleep apnea, such as sleep fragmentation and intermittent hypoxemia, are increasingly being implicated as risk factors for metabolic dysfunction [5] and are probably involved in the development of type 2 diabetes (T2DM), according to an increasing amount of sleep laboratory evidence [6]. Findings from these previous studies demonstrated that inadequate sleep, including the deprivation of absolute sleep duration and poor sleep quality, was associated with a higher risk of T2DM and obesity [7,8]. However, the number of hours of habitual sleep is associated with the risk of developing T2DM, and some studies show a U-shaped association has not been determined [9,10], whereas others report a uniform relationship between sleep duration and the risk of T2DM [11,12]. Moreover, although some studies suggested that insulin resistance caused by sleep abnormalities increases blood glucose levels, the exact mechanism of developing diabetes caused by sleep disorders remains unknown [13,14].

In this study, we assessed whether sleep duration was associated with incident T2DM in the healthy Korean population using large population-based 14-year longitudinal cohort data. We also examined the relationship between sleep quality and the risk of incident T2DM among study participants. We then sought to determine the mechanism involved in the development of T2DM in sleep disorders.

## 2. Materials and Methods

### 2.1. Study Population

The study population consisted of participants in the ongoing Korean Genome and Epidemiology Study (KoGES) Ansung and Ansan Cohort Study, a rural, community-based prospective study. This cohort study began in 2001 and included biennial follow-up examinations [15]. A total of 10,030 participants aged 40–69 years were enrolled at baseline (17,18). Participants who participated in this study were followed consistently from baseline (2001–2002) through the seventh biennial visit (2013–2014). We excluded subjects with no data on sleep duration or ESS (*n* = 531), fasting plasma glucose level (FPG, *n* = 267), and those already diagnosed with DM or taking antihyperglycemic medications (*n* = 416) at baseline. Finally, 8816 participants (4631 men and 4185 women) were enrolled in this study. Written informed consent was obtained from all participants at each visit. This study was approved by the Institutional Review Board of Hallym University Sacred Heart Hospital (IRB No. HALLYM 2021-08-002).

### 2.2. Data Collection

Participants in the KoGES Ansung and Ansan groups were examined. At each examination, they were asked to complete questionnaires about their personal and family medical history, smoking habits, alcohol consumption, exercise status, and medication use. Height and body weight were measured using standard methods, and body mass index (BMI) was calculated as weight divided by height in meters squared (kg/m^2^). Blood pressure was measured by an average of three readings in the morning after at least 10 min of rest in a sitting position [15]. In this study, participants were categorized by smoking status (never, past, or current smoker), and regular physical activity was defined as moderate or vigorous physical activity lasting ≥30 min daily. Heavy alcohol consumption was defined as alcohol consumption ≥30 g/day in men and ≥20 g/day in women.

Laboratory samples were obtained after a 12 h fast. Plasma samples were taken at 0 and 120 min during a 75 g oral glucose tolerance test (OGTT). Plasma glucose and insulin concentrations were determined by the hexokinase method and a radioimmunoassay kit (LINCO Research Inc., St. Charles, MO, USA), respectively. The Homeostasis Model Assessment of Insulin Resistance (HOMA-IR) index was calculated according to the following formula: (fasting plasma insulin concentration [µIU/mL] × FPG concentration [mg/dL]/405). Homeostasis model assessment of ß-cell function (HOMA-ß) was calculated as follows: 360 × fasting insulin (µU/mL)/FPG [mg/dL] − 63). The insulin glycogenic index (IGI) was calculated as follows: (60–0 min insulin [IU/mL]/60–0 min plasma glucose [mg/dL]). The Matsuda insulin sensitivity index was calculated as follows: (10,000/FPG [mg/dL] × fasting insulin (µU/mL) × mean plasma glucose [mg/dL] × mean insulin [µU/mL]). Total cholesterol, triglycerides, and high-density lipoprotein cholesterol were determined using a Hitachi 747 chemistry analyzer (Hitachi Ltd., Tokyo, Japan). Low-density lipoprotein cholesterol levels were determined using the Friedewald equation [16]. HbA1c was measured by high-performance liquid chromatography (Variant II; Bio-Rad, Hercules, CA, USA).

### 2.3. Sleep Duration and Quality of Sleep

Data on self-reported sleep duration and quality were collected through personal interviews at baseline. Sleep duration was assessed with the question, “On average, how many hours of sleep did you get per day during the past year (including nap time)?” [17]. Sleep duration was categorized into four groups: ≤5, 6–7, 8–9, and 10 ≥ h/day. For individuals with sleep duration of <10 h/day, sleep quality was assessed using the ESS, which measures excessive daytime sleepiness (EDS). The ESS asks respondents to rate their tendency to fall asleep during eight daily activities to determine their level of daytime sleepiness. EDS was defined as an ESS score > 10 [18].

### 2.4. Definition of the Disease

The primary endpoint in this study was incident diabetes during the follow-up period (from 2003 to 2014), defined as current use of oral glucose-lowering medication or insulin, a fasting glucose level ≥126 mg/dL (7.0 mmol/L), or a 2 h plasma glucose level ≥200 mg/dL (11.1 mmol/L) on the biennial OGTT, based on the diagnostic criteria suggested by the American Diabetes Association (ADA) [19]. Hypertension was defined as any of the following: (1) self-reported history of hypertension, (2) systolic or diastolic blood pressure ≥ 140/90 mmHg, or (3) use of antihypertensive medications [20]. Hyperlipidemia was defined as any of the following conditions: (1) self-reported history of lipid abnormalities, (2) total cholesterol level ≥ 240 mg/dL, or (3) use of lipid-lowering medications [21].

### 2.5. Statistical Analysis

General characteristics of study participants are reported as mean ± standard deviation for continuous variables and as numbers (percentages) for categorical variables. Group differences were tested with the unpaired Student’s *t* test for continuous variables and the χ^2^ test for categorical variables. HRs with 95% confidence intervals (CI) for risk of diabetes according to sleep duration and sleep quality index (ESS) were analyzed with multivariable Cox proportional hazard models. A restricted cubic spline transformation of sleep duration was used to assess nonlinear associations between sleep duration and diabetes risk. Statistical significance was set at *p* < 0.05. Statistical analysis was performed using SAS 9.4 (SAS Institute Inc., Cary, NC, USA) and R 3.1.0 (R Foundation for Statistical Computing, Vienna, Austria).

## 3. Results

### 3.1. Baseline Characteristics of the Study Participants

Baseline characteristics of study participants according to incident DM during the follow-up period are shown in Table 1. 

Of the 8816 participants, 1630 (22.7%) participants developed DM during the 14-year follow-up period. Participants who developed diabetes were older (51.49 ± 8.85 years in non-incident diabetes vs. 53.14 ± 8.65 years in incident diabetes). Women, smokers, and participants with a family history of diabetes, hypertension, and hyperlipidemia were more likely to be in the group that developed diabetes. BMI, systolic and diastolic blood pressure, total cholesterol, and triglyceride levels were higher at baseline in the group with incident diabetes. HbA1c levels were also higher at baseline in the group with incident diabetes (5.95 ± 0.61% in incident diabetes vs. 5.54 ± 0.40 in non-incident diabetes, *p* value < 0.001). Other glycemic variables, such as fasting glucose, post-load glucose, fasting insulin, post-load insulin, and HOMA-IR, were also higher in the incident diabetes group. However, both HOMA-B and IGI, indices of insulin secretory function, were lower in the group with incident diabetes. More than half of the participants had a sleep duration of <8 h/day in both groups. Patients with incident diabetes had a higher prevalence of sleep durations ≤5 h/day and 10 ≥ h/day (*p* value = 0.0078), and they also showed a tendency toward a higher prevalence of ESS scores > 10 than those who did not develop diabetes (13.15% in the group with non-incident diabetes vs. 14.79% in the group with incident diabetes, *p* value = 0.0808).

### 3.2. Association of Sleep Duration and Incident DM

Figure 1 shows the U-shaped relationship between sleep duration and the risk of incident diabetes in the study population. Both short and long sleep durations were significantly associated with a higher risk of diabetes. 

Table 2 shows the HRs for incident diabetes risk according to sleep duration. The incidence rate of diabetes was highest for long sleep durations (≥10 h per day). 

In multivariate analysis, after adjustment for age and sex (model 1), the HRs for sleep durations ≥10 h/day and ≤5 h/day were 1.65 (95% CI, 1.25–2.17) and 1.11 (95% CI, 0.97–1.27), respectively, compared with the reference group (sleep durations for the group with 6 to less than 7 h). Sleep durations of 8–9 h/day had the lowest risk of diabetes (HRs, 0.99; 95% CI, 0.83–1.05). These trends persisted after further adjustment for regular exercise, alcohol consumption, smoking, and family history of diabetes (model 2) and even after further adjustment for waist circumference, triglycerides, hypertension, and hyperlipidemia (model 3). In the fully adjusted model, which was additionally adjusted for fasting glucose and 2 h post-load glucose (model 4), the highest HRs were still significant for sleep durations of 10 h or more per day (HRs of 1.47 [95% CI, 1.12–1.94]). Sleep durations of ≤ 5h showed a trend toward higher diabetes risk compared with the reference group (HRs of 1.09 [95% CI, 0.95–1.26]).

### 3.3. Trends of Glycometabolic Variables for 14 Years and Risk of Incident DM

Figure 2 shows longitudinal changes in glycometabolic variables at each visit according to the daily sleep durations of study participants.

During the 14 years of follow-up, IGI, the index of insulin secretory function, decreased remarkably with long sleep durations (sleep duration ≥ 10 h) during the follow-up period. While HOMA-IR, the index of insulin resistance, slightly increased in this group during the same period. Among glycemic parameters, the increase in 1 h post-load glucose was most striking during long sleep durations compared with other parameters such as fasting glucose, 2 h post-load glucose, and HbA1c. The BMI of participants in the entire cohort was similar and remained stable in each sleep duration group over 14 years. This finding suggests that the increased risk of incident DM in individuals with long sleep durations is not caused by an increase in BMI due to less physical activity. Moreover, the increased risk of DM with long sleep duration is associated with decreased insulin secretion rather than increased insulin resistance.

### 3.4. Incident DM Risk and Quality of Sleep

We analyzed the risk of DM in participants whose sleep durations were less than 10 h to investigate whether sleep quality was related to the risk of DM, even in patients with similar sleep durations (Table 3). 

The quality of sleep of study participants was assessed at ESS. After adjustment for age and sex, individuals with EDS were associated with an increased risk of DM (HR, 1.39; 95% CI, 1.06–1.82) (model 1) compared with individuals without EDS. These trends persisted even after adjustment for regular exercise, alcohol consumption, smoking, family history of diabetes (model 2), and further adjustment for waist circumference, triglycerides, hypertension, and hyperlipidemia (model 3). In the fully adjusted model, which included adjustment for fasting glucose and 2-hr post-load glucose levels at baseline (model 4), patients with EDS also showed a higher risk of DM (HRs 1.52 [95% CI, 1.16–1.20]) compared to those without EDS.

## 4. Discussion

This was a community-based longitudinal cohort study in which participants were followed up for 14 years. In this study, we found that the association between sleep duration and incident DM was U-shaped; both short (≤5 h) and long (≥10 h) sleep durations were associated with an increased risk for the occurrence of incident DM. Compared with short sleep durations, long sleep durations were associated with a higher risk of incident DM. By analyzing trends in changes in glycometabolic variables over a 14-year period, we found that the increased risk of diabetes in the long sleep duration group may be caused by increasing glucose levels accompanied by decreased insulin secretion. Additionally, we found that EDS, an indicator of poor sleep quality (ESS > 10), was associated with an increased risk of DM. This implies that both sleep duration and sleep quality may be important for glucose metabolism. To our knowledge, this is the first longitudinal cohort study to investigate the trajectories of β-cell function and insulin resistance in relation to sleep duration over a 14-year period.

In this community-based longitudinal cohort study, we found a U-shaped association between sleep duration and risk of DM, with the lowest risk of type 2 DM when sleep duration was 8–9 h per day. A long sleep duration (≥10 h) showed a 47% higher risk of developing DM compared with a normal sleep duration (6–7 h). Moreover, short sleep duration tended to be associated with a higher risk of type 2 DM. Consistent with our findings, previous studies have indicated that both short and long sleep durations are associated with an increased risk of type 2 DM. Several studies have reported an association between sleep duration and the development of DM. A prospective study conducted among women in the Nurses Health Study showed that both short and long sleep durations were associated with an increased risk of developing DM [22]. After adjustment for BMI, this relationship remained only for late sleepers [22]. Another study from Germany demonstrated a significant positive association between difficulty maintaining sleep and the occurrence of incident type 2 DM in healthy individuals [23]. The Massachusetts Male Aging Study reported that a sleep duration of less than 6 h was associated with twice the risk of developing DM, even after adjusting for confounding factors in men without diabetes [24]. In most studies, a U-shaped relationship was observed between sleep duration and the risk of DM [9,10,25], while some reports showed that only short sleep duration was associated with a higher incidence of DM [11,12]. However, they did not examine changes in various glycometabolic parameters (e.g., fasting glucose, post-load glucose, insulin resistance, and insulin secretion markers) with sleep duration over 14 years. The pattern of changes in various glycemic parameters may provide clues to the mechanism underlying the association between sleep duration and incident diabetes.

To determine the possible mechanisms of this relationship, we examined the longitudinal changes in various metabolic parameters with sleep duration. In this study, insulin secretory function decreased significantly with longer sleep duration over the 14 years. Of note, in the biennial visits, fasting glucose was consistently higher with long sleep duration and, in particular, 1 h glucose increased in association with a decrease in insulin secretory function. Although insulin resistance also increased with long sleep duration, this trend was also consistently observed in the other sleep duration groups. Considering that the main pathogenesis of type 2 DM is insulin resistance and impaired insulin secretion, we hypothesize that the increased risk of DM during long sleep duration may be due to the deterioration of pancreatic beta cell function caused by excessive sleepiness [26,27,28,29]. Several previous cross-sectional studies have reported an association between sleep duration and diabetes-related markers. A cross-sectional study conducted in Canada found an increased risk of developing diabetes in individuals who slept less than 7 h (OR 1.58, 95% CI 1.13–2.31) [30]. The study showed that fasting glucose, fasting plasma insulin, and HOMA-IR were higher in short sleepers [30]. Another cross-sectional study in patients with type 2 DM found that poor sleep quality and inadequate sleep were associated with higher HbA1c levels [31]. In another report of middle-aged Caucasian participants who were generally healthy, poor sleep quality was significantly related to metabolic syndrome, and a significant association was found between sleep quality and fasting glucose, insulin, and insulin resistance [32].

Possible mechanisms for the association between sleep duration, sleep quality, and diabetes have been primarily explained by insulin resistance. A study conducted on adolescents with shorter and longer sleep durations found 20% higher levels of HOMA-IR compared to adolescents with average sleep durations [33]. Furthermore, Kim et al. [34]. reported a U-shaped trend at HOMA-IR according to sleep duration, which remained significant only in the group with longer sleep durations after adjustment for waist circumference. They suggested that the association between short sleepers and HOMA-IR may be primarily mediated through pathways related to central obesity, whereas the relationship between long sleepers and HOMA-IR appeared to be through pathways independent of central obesity [34]. Short sleep duration has been associated with metabolic syndrome and obesity. Restricted sleep duration reduces physical activity and energy levels [35,36], which could reduce activity-related energy expenditure and alter neuroendocrine control of appetite and hormone levels, such as reduced leptin and increased ghrelin levels [37]. These changes can lead to weight gain, obesity, and secondary insulin resistance [13,38]. However, our study showed that BMI hardly changed during the follow-up period in all sleep duration subgroups. Longer sleep duration might be associated with low levels of physical activity and low socioeconomic status [39,40]. Furthermore, activation of pro-inflammatory cytokines [41] and the sympathetic nervous system, as well as an increase in oxidative stress associated with sleep disturbances [42], leads to pancreatic beta cell death. In vitro cellular experiments also suggest that intermittent hypoxia associated with poor sleep quality decreases glucose-induced insulin secretion due to the downregulation of CD38 gene transcription, which is involved in Ca^2+^ mobilization and, thus, insulin secretion [43]. Our finding also adds some evidence to the hypothesis that sleep abnormalities may have a considerable role in the deterioration of pancreatic beta cells. 

The present study had several limitations. First, sleep duration was only recorded in the baseline data, and we could not collect data on changes in sleep duration or sleep patterns during the follow-up period due to a lack of data. Thus, it is possible that a single measure of exposure may not fully capture the lasting effects of sleep duration on long-term risk for type 2 DM. Although other variables besides sleep-related parameters may change during the follow-up period, we could not account for these possible changes in other variables over 14 years in our analysis. Additionally, due to the nature of the retrospective longitudinal cohort study, there were quite a lot of missing glycometabolic parameters among study participants during data collection for the follow-up period. Moreover, these analyses used self-reported sleep duration or quality, which is less reliable than objective measures such as polysomnography or actigraphy. As this cohort study was not originally designed to investigate sleep disorders, it does not collect data on objective measures of sleep duration and quality, and this is the critical limitation of our study. However, objective evaluation of sleep duration is not easily available in the population-based study due to high costs. Moreover, recent studies have shown that subjective reports of sleep duration based on questionnaires are moderately correlated with actigraphy-measured sleep. Therefore, plenty of epidemiologic studies used self-reported sleep data to obtain significant findings like our study. Second, different forms of sleep quality, such as sleep segments, were not considered. Finally, this study was based on a sample of Korean individuals in a population-based setting, which may limit the generalizability of our results to other ethnicities and settings.

## 5. Conclusions

Taken together, our results add to the evidence of an association between sleep duration, sleep quality, and diabetes. Long sleep duration and poor sleep quality in a population at risk of, but without, overt type 2 DM are demonstrated. This suggests that disruptions in sleep duration, particularly of 10 h or more, are a potential trigger for pancreatic beta cell damage, leading to impaired glucose metabolism. Sleep behavior is a potentially modifiable risk factor that could be considered in prevention strategies for type 2 DM. More studies are required to better understand the complex physiological interactions between sleep and glucose metabolism. This call for more research is supported by the potential public health benefits of considering sleep patterns in the prevention or management of chronic diseases such as diabetes.

## Figures and Tables

**Figure 1 jcm-12-02899-f001:**
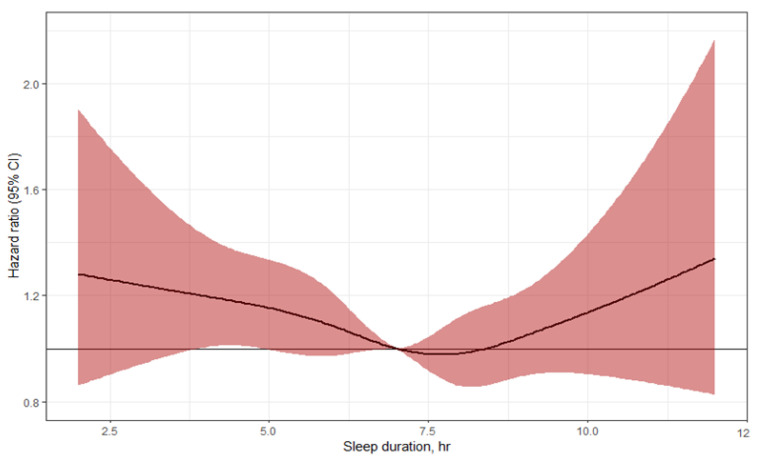
Multivariable-adjusted hazard ratios of sleep duration with risk of type 2 diabetes, evaluated by restricted cubic splines from Cox proportional hazards models. The models are adjusted for age, sex, regular exercise, alcohol consumption, smoking status, family history of diabetes, waist circumference, hypertension, hyperlipidemia, triglyceride, fasting plasma glucose, and 2 h post-load glucose.

**Figure 2 jcm-12-02899-f002:**
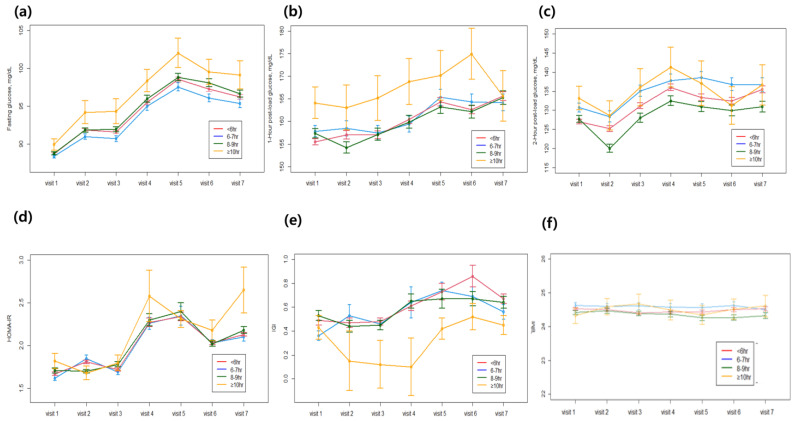
Longitudinal changes in glycometabolic variables according to the sleep duration: (**a**) fasting glucose, (**b**) post-load 1 h glucose, (**c**) post-load 2 h glucose, (**d**) HOMA-IR, (**e**) insulinogenic index, (**f**) body mass index.

**Table 1 jcm-12-02899-t001:** Baseline characteristics according to incident diabetes.

Variables	Incident Diabetes (−)	Incident Diabetes (+)	*p*-Value
	(*n* = 7186)	(*n* = 1630)	
Age (years)	51.49 ± 8.85	53.14 ± 8.65	<0.0001
Men	3862 (53.74)	769 (47.18)	<0.0001
Body mass index (kg/m^2^)	24.33 ± 3.02	25.35 ± 3.10	<0.0001
Systolic blood pressure (mmHg)	122.9 ± 18.38	128.7 ± 18.64	<0.0001
Diastolic blood pressure (mmHg)	81.91 ± 11.63	85.20 ± 11.25	<0.0001
Smoking status			<0.0001
current	1791 (25.13)	446 (27.62)	
past	1054 (14.79)	303 (18.76)	
never	4283 (60.09)	866 (53.62)	
Regular exercise	4035 (57.63)	896 (55.86)	0.195
Heavy alcohol drinker	4457 (63.58)	1006 (62.99)	0.6599
Daily alcohol consumption, g/day	19.60 ± 27.84	22.12 ± 27.69	0.0211
Family history of diabetes	709 (9.87)	256 (15.71)	<0.0001
Hypertension	1725 (24.01)	612 (37.55)	<0.0001
Hyperlipidemia	932 (12.97)	327 (20.06)	<0.0001
HbA1c (%)	5.54 ± 0.40	5.95 ± 0.61	<0.0001
Fasting glucose (mg/dL)	87.17 ± 8.52	95.29 ± 11.99	<0.0001
1 h post-load glucose (mg/dL)	147.4 ± 43.31	197.8 ± 45.36	<0.0001
2 h post-load glucose (mg/dL)	120.1 ± 35.07	163.3 ± 48.03	<0.0001
Fasting insulin (mU/L)	7.49 ± 4.73	8.00 ± 4.64	<0.0001
1 h post-load insulin (mU/L)	32.57 ± 32.22	32.60 ± 31.42	0.9705
2 h post-load insulin (mU/L)	27.68 ± 27.47	34.79 ± 32.93	<0.0001
HOMA-IR	1.63 ± 1.08	1.90 ± 1.16	<0.0001
Insulinogenic index	0.51 ± 2.45	0.34 ± 1.92	0.0028
Matsuda insulin sensitivity index	0.02 ± 0.19	0.02 ± 0.16	0.1155
Total cholesterol, mg/dL	196.7 ± 35.94	205.0 ± 35.65	<0.0001
HDL-C, mg/dL	50.29 ± 11.96	47.46 ± 11.32	<0.0001
Triglycerides, mg/dL	140.5 ± 95.42	185.8 ± 128.2	<0.0001
ESS > 10 score	945 (13.15)	241 (14.79)	0.0808
Sleep duration			0.0078
≤5 h	1167 (16.24)	283 (17.36)	
6–7 h	3989 (55.51)	900 (55.21)	
8–9 h	1873 (26.06)	391 (23.99)	
≥10 h	157 (2.18)	56 (3.44)	

HOMA-IR, homeostasis model assessment for insulin resistance; HDL-C, high-density lipoprotein cholesterol; ESS, Epworth Sleepiness Scale (ESS).

**Table 2 jcm-12-02899-t002:** Association of daily sleep duration with the incidence of diabetes.

	Sleep Duration (Hour)			
	≤5 h	6–7 h	8–9 h	≥10 h
Incident diabetes, case	283 (19.5%)	900 (18.4%)	391 (17.3%)	56 (26.3%)
Crude Hazard ratio	1.11 (0.97–1.27)	1	0.99 (0.87–1.11)	1.65 (1.25–2.17)
Model 1	1.13 (0.98–1.29)	1	0.93 (0.83–1.05)	1.51 (1.15–2.00)
Model 2	1.10 (0.96–1.26)	1	0.93 (0.83–1.05)	1.49 (1.13–1.96)
Model 3	1.10 (0.95–1.26)	1	0.93 (0.82–1.05)	1.47 (1.12–1.94)
Model 4	1.09 (0.95–1.26)	1	0.93 (0.82–1.05)	1.47 (1.12–1.94)

Model 1: adjusted for age and sex; Model 2: Model 1 plus regular exercise, alcohol consumption, smoking status, and family history of diabetes; Model 3: Model 2 plus waist circumference, hypertension, hyperlipidemia, and triglyceride; and Model 4: Model 3 plus fasting plasma glucose and 2 h post-load glucose.

**Table 3 jcm-12-02899-t003:** DM risk according to the presence of excessive daytime sleepiness (EDS) in patients who sleep less than 10 h per day.

	Excessive Day Time Sleepiness (EDS)	
	EDS (−)	EDS (+)
Incident diabetes, case	323 (16.4%)	68 (22.9%)
Crude Hazard ratio	1 (reference)	1.43 (1.09–1.86)
Model 1	1 (reference)	1.39 (1.06–1.82)
Model 2	1 (reference)	1.39 (1.06–1.82)
Model 3	1 (reference)	1.36 (1.04–1.78)
Model 4	1 (reference)	1.52 (1.16–1.20)

Model 1: adjusted for age and sex; Model 2: Model 1 plus regular exercise, alcohol consumption, smoking status, and family history of diabetes; Model 3: Model 2 plus waist circumference, hypertension, hyperlipidemia, and triglyceride; Model 4: Model 3 plus fasting plasma glucose, and 2 h post-load glucose.

## Data Availability

The data presented in this study are available on request from the corresponding author. The data are not publicly available due to the policy of Korea National health insurance system.

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
