# Peer review of "Association between Sleep Duration and Incident Diabetes Mellitus in Healthy Subjects: A 14-Year Longitudinal Cohort Study"

_jcm, 2023, doi:10.3390/jcm12082899_

Round 1

Reviewer 1 Report

The authors present a prospective, long-term cohort study aiming to assess the association between sleep duration and T2DM incidence (and its risk factors) in health Korean population. The analysis and results are presented clearly. I have the following comments that I hope the authors could address.

My major concern is the design of the study, where sleep duration and quality of sleep is only collected once through interview when the study is conducted over 14 years period. Is the objective of the present study in the manuscript a part of the original study design of the prospective Korean Genome and 62 Epidemiology Study (KoGES)-Ansung and Ansan Cohort Study? If so, the change of sleep duration and quality of sleep should be also considered, in my opinion, instead of a single measurement through interview. This is a longitudinal study, however, only glycometabolic variables are studied over time. The incident diabetes mellitus could happen any time over the 14 years, while the sleep duration and quality could change over time and before/after the diagnosis of diabetes as well.

Also, the study time of the interview to collect sleep data is not clear. Please provide more clarity in the visit schedule and when the key measurements (sleep duration, ESS, incident diabetes mellitus) are identified.

Minor typo: on page 8, line 58, “greater than 14 years”. I think the authors meant “over 14 years period”.

Reviewer 2 Report

The title of the manuscript ‘Association between Sleep Duration and Incident Diabetes Mellitus in healthy subjects: A 14-year longitudinal cohort study’ , the methods and results promised a unique longitudinal report from a nationally representative large cohort.

Specific comments:

(1)   The exclusion criterion was for those already identified to have diabetes (line nos 67-69). In the results a comparison was made between Incident DM+ in Table 1, line 134. Please reconcile the two

(2)   If the sleep data were captured only at baseline, this fact should be mentioned in the methods section (line nos 102,103)

(3)   Besides, there were no quantitative measures of sleep duration

(4)   Line 111: ‘Incident diabetes was defined..’: at baseline or on follow up? See comment (1) above

(5)   Line 188.189: ‘During the 14 years of follow-up, fasting glucose,1-h post-load glucose, 2-h post-load glucose and HOMA-IR showed increasing trends in all sleep duration groups.’: Did this persist even after accounting for other variables?

(6)   Line 227: ‘the increased risk of diabetes in the long sleep duration group may be caused by decreased insulin secretion increasing glucose levels after exercise.’: How does ‘after exercise” fit into the statement?

(7)   Line 250,251: ‘while some reports showed that short sleep duration was associated only with a higher incidence of DM’: Do the authors mean ‘'only short sleep'?’

(8)   ‘sleep duration greater than 14 years’: is ‘years’ a typo?

(9)   ‘..risk of DM during long sleep duration may be due to the deterioration of pancreatic beta cell function caused by excessive sleepiness’: Is there any evidence from previous studies? If yes, please cite

(10) These are only minor issues in an otherwise flawed design
